# Calibrated Uncertainty Quantification for Operator Learning via Conformal Prediction

**Ziqi Ma**                                                    *ziqima@caltech.edu*
*California Institute of Technology*

**David Pitt**                                                 *dpitt@caltech.edu*
*California Institute of Technology*

**Kamyar Azizzadenesheli**                                     *kamyara@nvidia.com*
*NVIDIA*

**Anima Anandkumar**                                           *anima@caltech.edu*
*California Institute of Technology*

**Reviewed on OpenReview:** *https://openreview.net/forum?id=cGpegxy12T*

## Abstract

Operator learning has been increasingly adopted in scientific and engineering applications, many of which require calibrated uncertainty quantification. Since the output of operator learning is a continuous function, quantifying uncertainty simultaneously at all points in the domain is challenging. Current methods consider calibration at a single point or over one scalar function or make strong assumptions such as Gaussianity. We propose a risk-controlling quantile neural operator, a distribution-free, finite-sample functional calibration conformal prediction method. We provide a theoretical calibration guarantee on the coverage rate, defined as the expected percentage of points on the function domain whose true value lies within the predicted uncertainty ball. Empirical results on a 2D Darcy flow and a 3D car surface pressure prediction task validate our theoretical results, demonstrating calibrated coverage and efficient uncertainty bands outperforming baseline methods. In particular, on the 3D problem, our method is the only one that meets the target calibration percentage (percentage of test samples for which the uncertainty estimates are calibrated) of 98%. Code is available at `https://github.com/neuraloperator/neuraloperator/tree/main` (UQNO module).

## 1 Introduction

Neural operators have been increasingly adopted to solve partial differential equations (PDE), demonstrating a significant speedup over traditional numerical methods. Their applications span various scientific and engineering domains, including weather forecasting (Pathak et al., 2022), carbon capture (Wen et al., 2023), automotive design (Li et al., 2023), and nuclear fusion (Gopakumar et al., 2023) to name a few. One key challenge in their adoption in real-world scenarios lies in uncertainty quantification—the capability to estimate how uncertain the model output is. Applications like plasma evolution prediction for nuclear fusion (Gopakumar et al., 2023) and pressure field prediction for automotive design (Li et al., 2023) are safety-critical and thus require reliable quantification of model uncertainty. Applications such as extreme weather forecasting Kurth et al. (2023); Lam et al. (2023); Persson & Grazzini (2007), carbon capture and storage (Wen et al., 2023) affect high-impact decision-making, which also requires uncertainty quantification. Existing methods for uncertainty quantification face challenges in three key areas, viz., lack of **function-space coverage**, **distribution-free calibration guarantee**, and **scalability**.

**Function-space formulation:** Neural operators provide function-space solutions that can be evaluated at any point in the domain. The necessary formulation of uncertainty for neural operators requires to be on

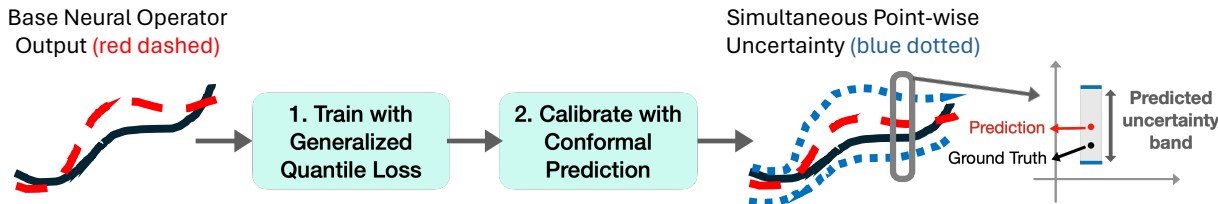

Figure 1: Overall schematic of UQNO, a risk-controlling quantile neural operator. In operator learning, the learned neural operator outputs a function (red dots sampled at grid points). We train a residual operator with generalized quantile loss and then calibrate with conformal prediction, which yields simultaneous pointwise uncertainty estimates with a PAC guarantee on calibration coverage—the expected percentage of true value (black) that lies within our predicted uncertainty bands (green=upper bound and yellow=lower bound). In this example, since the output is 1D, we output a pointwise uncertainty band. In higher dimensions, we output a pointwise heterogeneous uncertainty ball.

the function space, which provides simultaneous uncertainty estimation for all points. For example, in an automotive design setting that uses neural operators to predict surface pressure Li et al. (2023), simultaneous uncertainty estimates on the whole surface can inform designers of structured error around the front face ridge, as shown in Figure 1, whereas a single-point or aggregate measure of uncertainty cannot convey such information.

Uncertainty quantification in function spaces is more challenging than naively combining point predictions in the function space. Even if we obtain point-wise guarantees on calibration with probability $1 - p$, the standard union bound leads to a loose bound on the probability of simultaneous calibrations. Prior work focuses on single-point uncertainty estimation (Guo et al., 2023), and theoretical developments investigate uncertainty quantification in the function space, yet focus on transformed formulations such as functional projection or scalar function properties such as pseudo-density or loss (Lei et al., 2015; Benitez et al., 2023).

**Distribution-free calibration guarantee:** Classical deep learning uncertainty quantification methods such as MCDropout (Gal & Ghahramani, 2016) or ensemble learning (Maddox et al., 2019) do not provide calibration guarantees. These methods output mean and variance estimates under Gaussian assumptions, and the results are generally heuristic. Recent works on uncertainty quantification for operator learning are either heuristic (Guo et al., 2023; Akhare et al., 2023; Nehme et al., 2023), or rely on Gaussian assumptions and approximations that may not hold in real-world settings (Magnani et al., 2022). Heuristic uncertainty estimation is insufficient for safety-critical or high-impact applications. We need a method with a rigorous calibration guarantee that works "in the wild", where distributional assumptions might be broken. Since real-world applications have finite data, we also want our calibration guarantee to be finite-sample rather than asymptotic.

**Scalability:** Frameworks such as Bayesian inference are principled, yet challenging to scale up. For example, Meng et al. (2022) considers problems with per-function samples on the order of hundreds, yet many practical applications are magnitudes larger in scale, e.g., 7.2 million mesh points in fluid dynamics Li et al. (2023), creating computational challenges for such methods.

We present an uncertainty-quantified neural operator (UQNO) framework that simultaneously addresses these challenges by leveraging the conformal prediction framework. UQNO is a distribution-free and finite-sample functionally calibrated conformal prediction method built on the framework of risk-controlling quantile operator learning that leverages the classical conformal prediction principle (Lei et al., 2015; Angelopoulos & Bates, 2021). A high-level schematic of our method is shown in Figure 1. We develop a generalized quantile loss extended to the function space to train a neural operator that takes a function as input and outputs a heuristic uncertainty band that can be queried at any point. Then, we provide the conformal prediction

framework to calibrate the risk-controlling prediction set for operator learning. This framework enables function prediction along with point-wise calibrated coverage uncertainty.

We demonstrate that our method satisfies the desired calibration guarantee while providing efficient uncertainty bands, outperforming baselines in a 2D Darcy flow problem, and a 3D automotive surface pressure field prediction. In the 2D Darcy flow problem, our method provides uncertainty bands that are **1.52x** tighter than MCDropout and **76.1x** tighter than Laplace approximation while satisfying the target calibration percentage of 98%. In the 3D car surface pressure field prediction problem, our method is the **only** method that satisfies the target calibration percentage of 98%. Our main contributions are as follows:

- We provide a function-space uncertainty formulation for operator learning by leveraging the conformal prediction framework.

- We provide calibrated uncertainty estimates simultaneously for all points on the function domain.

- We provide a PAC bound on the coverage percentage of our uncertainty estimates.

We empirically demonstrate that UQNO outperforms existing uncertainty methods in terms of calibration percentage and band tightness for both data-rich and data-scarce regimes, with per-function point samples up to $177k$.

## 2 Problem Formulation

### 2.1 Uncertainty Quantification for Operator Learning

Operator learning Li et al. (2020) can be formulated as,

$$\hat{\mathcal{G}} = \min_{\mathcal{G}} \mathbb{E}_{(a,u)} l(\mathcal{G}(a), u) \tag{1}$$

where $a \in \mathbb{A} = \mathbb{A}(D, \mathbb{R}^{d_a})$ is the input function, $u \in \mathbb{U} = \mathbb{U}(D, \mathbb{R}^{d_u})$ is the output function, $D \subset \mathbb{R}^d$ is compact and $\mathbb{A}, \mathbb{U}$ are separable Banach spaces. We aim to learn the operator $\hat{\mathcal{G}}$ which minimizes expected loss $l : \mathbb{U} \times \mathbb{U} \to \mathbb{R}$ over the distribution of input-output functions. The point estimate of operator $\mathcal{G} : \mathbb{A} \to \mathbb{U}$ does not give information of uncertainty beyond its empirical loss on available data. With uncertainty quantification, we want to output an uncertainty set instead of a point estimate. We formulate the uncertainty quantification problem as finding

$$C : \mathbb{A} \to \mathbb{U}', \quad \mathbb{U}' = \mathbb{U}'(D, \mathcal{B}(\mathbb{R}^{d_u})) \tag{2}$$

where $\mathcal{B}(\mathbb{R}^{d_u})$ is the Borel set on $\mathbb{R}^{d_u}$, i.e., $C$ evaluated at any function $a \in \mathbb{A}$ gives a correspondence (Aliprantis & Border, 2006) $(D, \mathcal{B}(\mathbb{R}^{d_u}), C(a))$ which maps any point $x \in D$ to a prediction set on $\mathbb{R}^{d_u}$, the co-domain of $u$.

### 2.2 Risk-Controlling Prediction Set for Operator Learning

The uncertainty mapping $C$ defined above is required to be calibrated following the established definition of risk-controlling prediction set Angelopoulos et al. (2022); Bates et al. (2021). The Risk-controlling prediction set for operator learning is defined as the following PAC bound: A $(\alpha, \delta)$-risk-controlling prediction set satisfies

$$\mathbb{P}_{(u,a)}[\mathbb{E}_x[\mathbb{1}\{u(a)(x) \in C_\lambda(a)(x)\}] < 1 - \alpha] \leq \delta \tag{3}$$

where the predicted uncertainty set $C_\lambda(a)(x)$ is connected. The risk-controlling prediction set specifies that the probability (over $(u, a)$) that when sampling points in its domain, the expected percentage of "violating points" (i.e. points whose true function value falls outside of the provided uncertainty ball) exceeds level $\alpha$ does not exceed $\delta$.

### 2.3 Parameterizing Uncertainty Set

We parameterize the uncertainty set as a pointwise ball on the co-domain of output function u. Assuming we have a non-calibrated uncertainty estimating operator $E$ with $E(a)(x) \in \mathbb{R}$ estimates the pointwise radius of a "uncertainty ball" for each new input function a, we specify the following parameterization of

C. We choose this formulation below because although obtaining calibrated uncertainty estimates is hard, we have access to heuristic uncertainty estimates methods that we extend to operator learning—in our case, we develop a "residual neural operator" for its flexibility and discretization convergence. Defining the calibrated uncertainty set as a scaled version of the heuristics is a way to transform the function-space uncertainty quantification problem into a problem on scalers, thus making it amenable to existing principles from conformal prediction.

$$C_\lambda(a)(x) = \{p \in \mathbb{R}^{d_u} \colon \|p - \hat{\mathcal{G}}(a)(x)\|_2 \le \lambda E(a)(x)\} \tag{4}$$

For a given function $a$ and given point $x$, the set defined above is the set of points in the co-domain of output function $u$ whose distance (measured by 2-norm) from the base operator prediction is no greater than $\lambda E(a)(x)$, i.e. the true value lies within a ball centered at the base operator prediction with a radius $E(a)(x)$ scaled by $\lambda$. In our implementation, $E$ is parameterized by a neural operator as described in Section 3.1.

## 3 Methods

To obtain a valid risk-controlling prediction set $C$, we leverage two main ingredients: 1) a generalized quantile loss formulation for operator learning to obtain $E$ in equation 4 that provides a good heuristic for model uncertainty; 2) the conformal prediction framework to select $\lambda$ so that the prediction set is well-calibrated.

### 3.1 Pre-Calibration Quantile Neural Operator

We use state-of-the-art neural operator architectures to parameterize $E$ in in equation 4—Fourier Neural Operator (FNO) (Li et al., 2021) for the 2D Darcy problem and its variant Geometric-Informed Neural Operator (Li et al., 2023) for the 3D pressure field prediction problem. We generalize quantile loss (Koenker, 2005; Chung et al., 2021) to the operator learning setting. Similar to the canonical quantile loss formulation, this loss penalizes out-of-band distance weighted by $1 - \alpha$ and in-band distance weighted by $\alpha$, and thus encourages the model to output a value close to the $1 - \alpha$ percentile of the error magnitude seen during training. $E$ is trained with the loss defined below:

$$L_\alpha(\hat{\mathcal{G}}(a), u) = \alpha \int_D \mathbb{1}\{\|u(x) - \hat{\mathcal{G}}(a)(x)\|_2 > E(a)(x)\}(\|u(m) - \hat{\mathcal{G}}(a)(m)\|_2 - E(a)(m))dx$$

$$+ (1-\alpha) \int_D \mathbb{1}\{\|u(x) - \hat{\mathcal{G}}(a)(x)\|_2 \le E(a)(x)\}(E(a)(x) - \|u(x) - \hat{\mathcal{G}}(a)(x)\|_2)dx \tag{5}$$

where $\hat{\mathcal{G}}$ is the base neural operator as defined in Equation equation 1. It is known that quantile loss does not provide well-calibrated quantiles (Chung et al., 2021), and we only use this prediction as a pre-calibration estimate.

### 3.2 Calibration via Split Conformal Prediction

We leverage conformal prediction as an overall framework (Vovk et al., 2005), which provides a finite-sample, distribution-free confidence set for a scalar-value prediction problem, utilizing calibration set that is exchangeable with test data. Split conformal prediction is used to avoid re-calibration for each new test data point Vovk et al. (2020). Note that we do not require training data to come from the same distribution.

Under this framework, given training set $D_{\text{train}} = \{(a_i^t, u_i^t)\}$, a calibration set $D_{\text{cal}} = \{(a_i^c, u_i^c)|i = 1, ...n\}$, and nonconformity score function $s : (\mathbb{A}, \mathbb{U}) \to \mathbb{R}$, under the assumption that the calibration set and test set are exchangeable, the possible outcomes which the $1 - \alpha$ quantile of $s_i$ obtained on the calibration set provides a valid estimate of the $1 - \alpha$ quantile on the test set, we have the following: Given training set $D_{\text{train}} = \{(a_i^t, u_i^t)\}$, calibration set $D_{\text{cal}} = \{(a_i^c, u_i^c)|i = 1, ...n\}$, we can define any nonconformity score function $s : (\mathbb{A}, \mathbb{U}) \to \mathbb{R}$ and calculate the score for all samples in the calibration set, i.e. $s_i = s(a_i^c, u_i^c)$. i.e., let

$$\hat{q} = \frac{\lceil (n+1)(1-\alpha) \rceil}{n} \text{'th quantile of } s_i$$

$$C(a) = \{u \in \mathbf{U} : s(a, u) \le \hat{q}\} \tag{6}$$

provides the guarantee that $P[u \in C(a)] \geq 1 - \alpha$ on a new test sample. We define the nonconformity score function $s : (\mathbb{A}, \mathbb{U}) \to \mathbb{R}$ as follows:

$$s_{\mathcal{G}}(a, u) = \sigma_{\lfloor 1 - \alpha + t \rfloor}, \tag{7}$$

where $t > \sqrt{-\frac{\ln \delta}{2m}}$, $\sigma_j$ is the $j$'th smallest value in the set $\{ \frac{\|u(x_i) - \hat{\mathcal{G}}(a)(x_i)\|_2}{E(a)(x_i)} | i = 1, ... m \}$, $m$ is the number of points on the domain for each function sample. Combined with equation 4, this gives a $(\alpha, \delta)$ risk-controlling confidence set. Theoretical derivation is presented in Section 5.

### 3.3 Risk-Controlling Quantile Neural Operator

Combining the two ingredients above, we obtain a risk-controlling quantile neural operator that can simultaneously predict uncertainty estimates for each point, given a test function. The overall algorithm is described in Algorithm 1.

---

**Algorithm 1:** Risk-Controlling Quantile Neural Operator

---

1: **Input:** Base model $\mathcal{G}$, training set (separate from the training set of $\mathcal{G}$) $D_{\text{train}'} = \{(a_i', u_i')\}$ , calibration set $D_{\text{cal}} = \{(a_i^c, u_i^c)\}$ of size n. Each $(a_i^c, u_i^c)$ is measured with discretization $m_i$ i.e. $a_i^c, u_i^c$ evaluated at $x_1, ... x_{m_i} \in \mathbb{R}^d$.

2: **Train:** Train quantile neural operator E on $D_{\text{train}'}$ with generalized quantile loss defined in equation 5

3: **Calibrate:**

4: **for** $(a_i^c, u_i^c) \in D_{\text{cal}}$ **do**

4: $\quad s(a_i^c, u_i^c) \leftarrow \sigma_{\lfloor} 1 - \alpha + t \rfloor$ as defined in equation 7 $t > \sqrt{-\frac{\ln \delta}{2\bar{m}}}$ where $\bar{m} = \min\{m_1, ..., m_n\}$

5: **end for**

6: $\hat{\lambda} \leftarrow 1 - \frac{\lceil (n+1)(\delta - e^{-2\bar{m}t^2}) \rceil}{n}$ quantile of $s(a_i^c, u_i^c)$ on $D_{\text{cal}}$

7: **Predict:** given a test input function $a$ and any $x \in \mathbb{R}^d$, output uncertainty ball $B(\mathcal{G}(a)(x), \hat{\lambda}E(a)(x))$, which is a $(\alpha, \delta)$ risk-controlling prediction equation 3

---

This method has three advantages: (1) **Heteroscedasticity**: The base uncertainty estimator is parametrized as a neural operator Li et al. (2021; 2023), which can predict higher uncertainty for input samples that are dissimilar to training input. (2) **Simultaneous pointwise prediction**: Simultaneous prediction of uncertainty estimates for all points on the domain allows for a structural understanding of error, as visualized in Figures 2 and 5. This is not possible in prior methods such as Guo et al. (2023) and Benitez et al. (2023). (3) **Controllablility**: We note that both the domain coverage threshold $\alpha$ and function-level coverage $\delta$ (equation 3) are user-specified. This is demonstrated empirically in Figures 3 and 4. We note that the calibration guarantee of our method is an upper bound over the true calibration, i.e., the actual coverage is guaranteed to be no less than the target coverage. The sparser our sample points are (on the domain), the more conservative our band becomes.

Our formulation naturally extends to scenarios where data is in various discretizations/irregular grid geometries. Different discretizations is regarded as different approximations of continuous functions that satisfy exchangeability. The theoretical derivation is presented in Section 5.2.

## 4 Experiments

We demonstrate our neural operator calibration method on two tasks: a high-resolution, data-rich 2D Darcy flow problem and a data-scarce pressure prediction problem on 3D car surfaces. We compare with existing methods that can be used or adapted for these tasks, including,

**MCDropout** Gal & Ghahramani (2016): which predicts uncertainty by aggregating results from multiple (we use 10) models trained with random dropout, which we generalize to the operator learning setting.

**Deep Ensemble** Lakshminarayanan et al. (2017): which predicts uncertainty by aggregating results from an ensemble (we use 10) of models with Gaussian assumptions, which we generalize to the operator learning setting.

Table 1: Calibration percentage and bandwidth comparison of UQNO with other methods (✓ indicate passing target of $1 - \alpha$, and ✗ indicate below target). We see that UQNO consistently provides the tightest uncertainty band that satisfies calibration conditions. For both tasks, we show a high-domain-threshold scenario ($\alpha = 0.02$ for Darcy and $\alpha = 0.04$ for car, $\alpha$ values larger for car due to its lower resolution due to the correction term t in Equation 7) and a low-domain-threshold scenario ($\alpha = 0.1$ for Darcy and $\alpha = 0.12$ for car). In the Darcy problem, where we have sufficient data, most methods satisfy our calibration target of 98%, but our method gives the most efficient band. In the car problem with scarce data, our method is the only method that provides calibrated results. Training time is approximate GPU hours on a single RTX4090.

| | 2D Darcy Flow | | | | | 3D Car Pressure | | | | |
| | High domain threshold | | Low domain threshold | | | High domain threshold | | Low domain threshold | | |
| | Bandwidth | Calibrated | Bandwidth | Calibrated | Training time | Bandwidth | Calibrated | Bandwidth | Calibrated | Training time |
|---|---|---|---|---|---|---|---|---|---|---|
| MC Dropout | 0.00108 | ✓ 99.5% | 0.00108 | ✓ 100% | 10 hours | 0.12332 | ✗ 18.9% | 0.12332 | ✗ 66.7% | 30 hours |
| Deep Ensemble | 0.00075 | ✗ 32.2% | 0.00075 | ✗ 90.2% | 10 hours | 0.42972 | ✗ 74.8% | 0.42972 | ✗ 96.4% | 30 hours |
| Laplace Approx. | 0.03453 | ✓ 100% | 0.03453 | ✓ 100% | 1 hour | 1.94708 | ✗ 94.6% | 1.94708 | ✗ 94.6% | 3 hours |
| Neural Posterior | N/A | ✗ 0% | N/A | ✗ 15.1% | 1 hour | N/A | ✗ 0% | N/A | ✗ 0% | 3 hours |
| UQNO | **0.00062** | ✓**98.6%** | **0.00045** | ✓ **99.8%** | **2 hours** | **0.55077** | ✓ **98.2%** | **0.40018** | ✓ **100%** | **6 hours** |

**Laplace approximation** Magnani et al. (2022): which takes point predictions as the Maximum a posteriori (MAP) estimator and does local Laplace approximation by leveraging the Hessian of the last layer weights. Note that bandwidth is not defined for this method since its predicted uncertainty subspace is unbounded.

**Neural posterior principal components** Nehme et al. (2023): which trains the model to not only predict a point estimate but also the first k principal components of residual, thus outputting an "uncertainty subspace" based on these components.

We focus on two aggregate metrics: (1) **Calibration percentage**: the percentage of functions in the test set that satisfy the target threshold. For each test sample, "satisfying the target threshold" means over $1 - \alpha$ of uniformly-sampled points on the function domain lie within our predicted uncertainty sets, as described in Equation 4. Note that calibration percentage is an aggregate metric defined on the whole test set. For a single instance, we also define "coverage" as the percentage of uniformly sampled points on the function domain that lie within predicted uncertainty sets for a specific function. This metric is used in single-instance visualizations, as shown in Figures 2, 5. (2) **Bandwidth**: the average predicted uncertainty ball radius averaged across all sampled points on the domain. This metric shows how efficient a method is—trivially, an infinitely large uncertainty prediction will always have perfect calibration percentage, yet is uninformative since the balls are not efficient.

We demonstrate that our method provides good calibration and tight bandwidth, compared to baselines. The Darcy flow task illustrates a data-rich setting with 5000 data points, whereas the car pressure prediction task represents a data-scarce setting with a 500-sample training set for a 3D problem. Table 1 provides statistics of bandwidth and calibration percentage percentage for all methods on both tasks. We see our method consistently outputs the most efficient uncertainty bands that satisfy the calibration targets. In the data-rich setting of Darcy flow, we achieve up to 1.53x efficiency improvement in band tightness. More notably, in the data-scarce problem of 3D car pressure prediction, all other baselines fail to provide calibrated uncertainty estimates in both high-domain-threshold and low-domain-threshold settings.

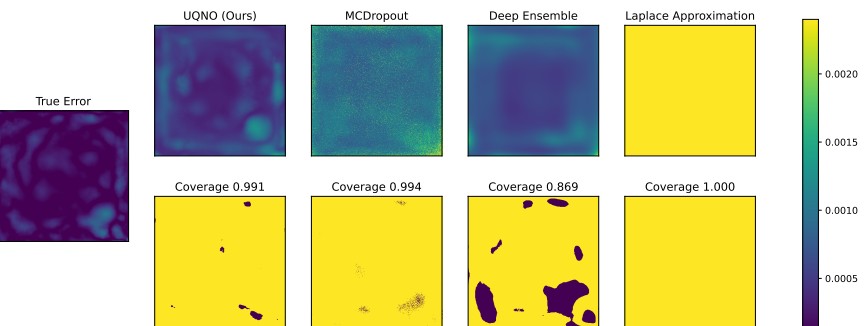

Figure 2: Uncertainty quantification comparison across methods on 2D Darcy flow problem. The leftmost heatmap plots true error. The top panels show the predicted pointwise uncertainty, and the bottom panels show the coverage (i.e. true error less than predicted error) for each point on the domain—yellow points are covered by our predicted uncertainty bands, and purple points are uncovered. The coverage percentage for each method is shown above the bottom panels. Our method of UQNO predicts uncertainty that corresponds well with true error while providing 99.1% domain coverage. MCDropout does not capture the uncertain region well. Laplace approximation greatly overestimates uncertainty—on the scale of $50\times$ larger than true error, and thus appears all yellow in the heatmap.

## 4.1 Darcy Flow

The 2D Darcy Flow problem is a second-order, linear, elliptic PDE of the form below:

$$-\nabla \cdot (a(x)\nabla u(x)) = f(x) \quad x = (0,1)^2$$
$$u(x) = 0 \quad x = \partial(0,1)^2 \tag{8}$$

This is a data-rich scenario with 5000 total training data and $421 \times 421$ resolution, for which we obtain the ground truth from prior work Li et al. (2021). We fix the same Fourier Neural Operator architecture for all methods. For UQNO, we split the training set in half for training the base and the quantile model. Figure 2 provides a visualization of domain-level uncertainty predictions and coverage for one function sample for UQNO (our method), MCDropout, and Laplace approximation. UQNO captures the error structure well and provides good domain-level coverage.

In addition to coverage, UQNO also provides a tight band, as shown in Figures 3 and 7. Among the methods that satisfy calibration (green shaded), our method (purple dot) consistently provides the lowest bandwidth. We also plot the average bandwidth against calibration percentage for different methods in Figure 3. UQNO provides the tightest band among methods while providing a calibration percentage of 98.8%. We note that the neural posterior principal components method does not provide a finite band since it outputs a subspace, and thus, bandwidth does not apply. Nevertheless, we take the percentage of error variance falling in the predicted subspace as its calibration percentage to compare with the other methods. We see the variance ratio falling into the dominant linear subspace is low (15%), suggesting the nonlinearity of the problem. We further demonstrate the flexibility of UQNO by showing calibration results at different target calibration percentages ($1-\delta$ in Equation 3) and domain coverage thresholds ($1-\alpha$ in Equation 3). There is a trade-off between the tightness of the uncertainty band and calibration percentage—intuitively, an infinitely large uncertainty set has a perfect calibration percentage yet is uninformative. By varying the domain threshold ($\alpha$ in Equation 3) and target percentage ($\delta$ in Equation 3), we calibrate UQNO to provide different guarantees. Figure 4 demonstrates the flexibility of our method by trading off band tightness with calibration percentage. Intuitively, wider bands correspond to a higher calibration percentage.

We also show the efficacy of the calibration percentage guarantee in Figure 6. The empirical results agree with our theoretical guarantee that the true calibration percentage should be no less than the expected calibration percentage of $1-\delta$ as in Equation 3 for various domain threshold $\alpha$ values. We note that by definition, our method is conservative bound due to the finite resolution of data samples (the t term in Equation 7).

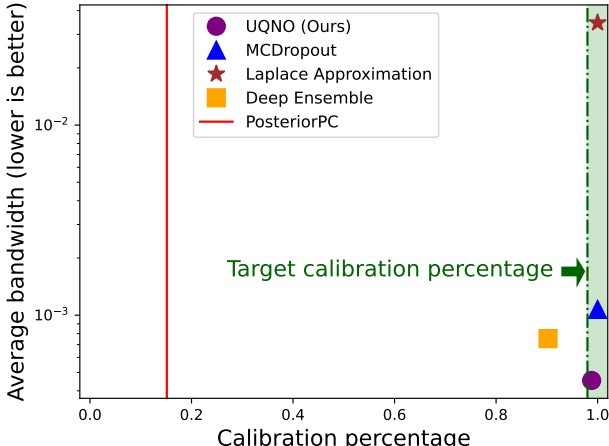

Figure 3: Bandwidth vs. calibration percentage comparison across methods on 2D Darcy flow problem. We see a clear advantage of UQNO (purple dot), providing $1.52\times$ tighter band than MCDropout and $76.1\times$ tighter band than Laplace approximation. This plot is generated with $\alpha = 0.1$, target calibration percentage 98% for UQNO, and $N = 3$ principal components for posterior principal component method.

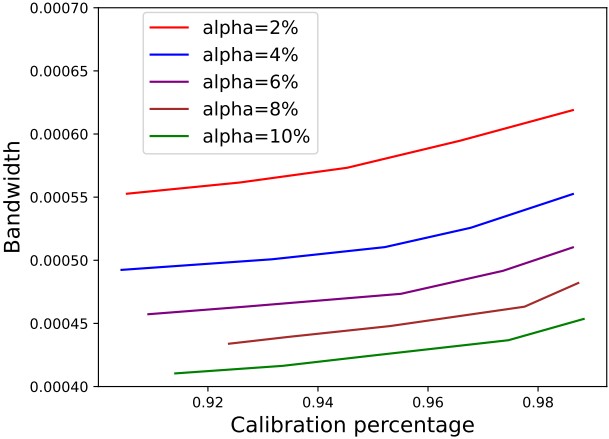

Figure 4: Bandwidth vs. calibration percentage trade-off of UQNO on 2D Darcy flow. Each curve shows the bandwidth vs. calibration percentage trade-off for a fixed domain threshold ($\alpha$ in Equation 3), demonstrating the flexibility to achieve a higher calibration percentage with wider bands. Overall bandwidth increases as the domain threshold becomes more stringent (smaller $\alpha$).

## 4.2  3D Car Surface Pressure Prediction

This task aims to learn the solution operator for 3D computational fluid dynamics (CFD) simulations of the Navier-Stokes equation for car shapes from Umetani & Bickel (2018) modified from the Shape-Net dataset Chang et al. (2015) car category. The car surface is represented as a 3D mesh of 3586 points, and we obtain ground-truth time-averaged pressure fields by solving the Reynolds-averaged Navier–Stokes (RANS) equation with a finite element solver Zienkiewicz et al. (2014) with Reynolds number $5\text{x}10^6$ and inlet velocity $72km/h$ as in Li et al. (2023). This is a data-scarce setting with only 500 total training samples on a 3D mesh of 3586 points. We have access to the signed distance function (SDF) and point-cloud representations as input and aim to predict the pressure at every mesh point on the car surface. Figure 5 compares UQNO, MCDropout, and Laplace approximation on a sample car shape and shows UQNO to capture the error structure and provides good coverage of 99.7%, albeit being conservative (over-estimates uncertainty).

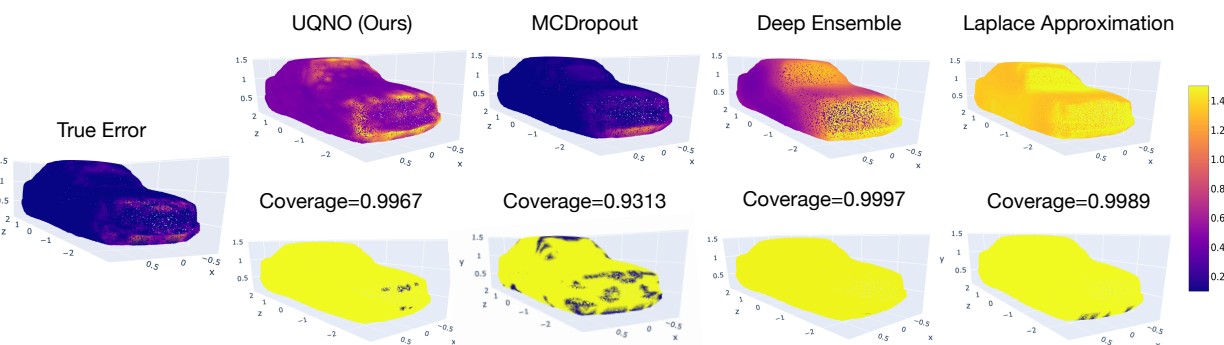

Figure 5: Uncertainty quantification comparison across methods on 3D car pressure prediction problem. The leftmost heatmap plots true error. The top panels show the predicted pointwise uncertainty, and the bottom panels show the coverage (i.e. true error less than predicted error) for each point on the domain—yellow points are covered by our predicted uncertainty bands, and purple points are uncovered. The coverage percentage for each method is shown above the bottom panels. Our method, although conservative (over-estimates uncertainty), is able to capture the error pattern on both the top and bottom of the front face of the car. Due to the conservative nature of our calibration procedure, our method satisfies the coverage threshold of $> 98\%$ (actual 99.7%). MCDropout, although on the same scale as true error, misses the top error region and fails to meet the coverage threshold. Laplace approximation greatly over-estimates the error and does not capture the error structure well.

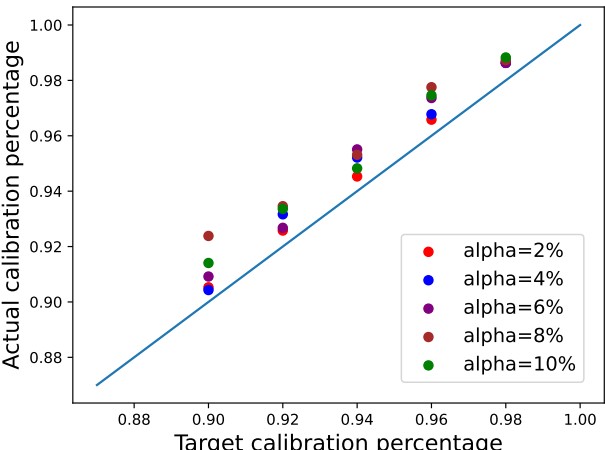

Figure 6: Actual vs. target calibration percentage of UQNO. Different $\alpha$ values are plotted with different colors. We see the guarantee is always satisfied.

We see that UQNO provides an interpretable error structure that shows high error along the edges of the front of the car, whereas MCDropout mainly captures the bottom region, and Laplace approximation provides a very coarse error estimate around the front center and over-estimates the error magnitude. Figure 7 compares the bandwidth and calibration percentage across different methods, showing UQNO (calibration percentage 100%) to be the only method that satisfies the calibration percentage target of 98%. We see that MCDropout has a low calibration percentage at 66.7%, and Laplace approximation gives a 4.85x wider band while only providing a 94.6% calibration percentage. The low calibration percentage of the Neural Posterior PC method (0%) shows the problem to be highly nonlinear.

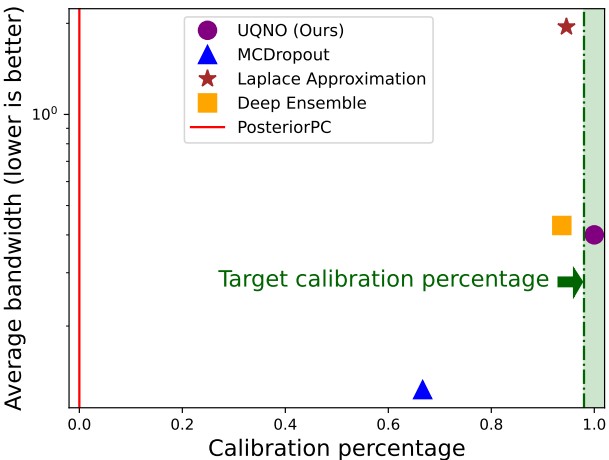

Figure 7: Bandwidth vs. calibration percentage comparison across methods on 3D car pressure prediction problem. Note the y-axis is plotted on a logarithmic scale. We see that UQNO (calibration percentage 100%) is the only method that meets the target of 98%. MCDropout gives a low calibration percentage of 66.7%, and Laplace approximation gives a 4.85x wider band while only providing 94.6% calibration percentage.

## 5 Theoretical Guarantee

Recall from Section 2.2, our goal is to quantify uncertainty for the operator learning problem 1:

$$\hat{\mathcal{G}} = \min_{\mathcal{G}} \mathbb{E}_{(a,u)} l(\mathcal{G}(a), u)$$

where $a \in \mathbb{A} = \mathbb{A}(D, \mathbb{R}^{d_a})$ is the input function, $u \in \mathbb{U} = \mathbb{U}(D, \mathbb{R}^{d_u})$ is the output function, $D \subset \mathbb{R}^d$ is compact and $\mathbb{A}, \mathbb{U}$ are separable Banach spaces. We formulate uncertainty quantification as finding a correspondence $C$:

$$C : \mathbb{A} \to \mathbb{U}', \quad \mathbb{U}' = \mathbb{U}'(D, \mathcal{B}(\mathbb{R}^{d_u})) \tag{9}$$

where $\mathcal{B}(\mathbb{R}^{d_u})$ is the Borel set on $\mathbb{R}^{d_u}$. $C$ evaluated at any function $a \in \mathbb{A}$ gives a correspondence (Aliprantis & Border, 2006) $(D, \mathcal{B}(\mathbb{R}^{d_u}), C(a))$ which maps any point $x \in D$ to a prediction set on $\mathbb{R}^{d_u}$, the co-domain of $u$. To obtain uncertainty quantification with conformal guarantee, we want to construct a $(\alpha, \delta)$-risk-controlling confidence set satisfies:

$$\mathbb{P}_{(u,a)}[\mathbb{E}_x[\mathbb{1}\{u(a)(x) \in C_\lambda(a)(x)\}] < 1 - \alpha] \leq \delta$$

where $C_\lambda(a)$ is the desired operator that outputs the radius of an "uncertainty ball" at any given point $x$ on the function domain.

$$C_\lambda(a)(x) = \{p \in \mathbb{R}^{d_u} : \|p - \hat{\mathcal{G}}(a)(x)\|_2 \leq \lambda E(a)(x)\}$$

with the minimum possible $\lambda$.

### 5.1 Fixed Discretization

We start with the simple case of fixed discretization where $m$ points are sampled from the domain for each function in calibration set. Following the definition of nonconformity score introduced in Equation 7,

$$s_t(a, u) = \sigma_{\lfloor 1 - \alpha + t \rfloor}$$

where $\sigma_j$ is the j'th smallest value in the set $\{ \frac{\|u(x_i) - \hat{\mathcal{G}}(a)(x_i)\|_2}{E(a)(x_i)} | i = 1, ...m \}$, $t > \sqrt{-\frac{\ln \delta}{2m}}$. We have,

$$s_t(a, u) > \lambda \Rightarrow \sum_{i=1}^{m} \mathbb{1} \{ \frac{\|u(x_i) - \hat{\mathcal{G}}(a)(x_i)\|_2}{E(a)(x_i)} < \lambda \} < m(1 - \alpha + t)$$

$$\Rightarrow \frac{1}{m} \sum_{i=1}^{m} \mathbb{1} \{ u(x_i) \in C_\lambda(a)(x_i) \} < 1 - \alpha + t. \tag{10}$$

Via conformal prediction (Equation 6), let $\hat{\lambda} = 1 - \frac{\lceil (n+1)(\delta - e^{-2mt^2}) \rceil}{n}$'th quantile of $s(a, u)$ on calibration set (of size $n$), so we know for test $(a, u)$, we obtain,

$$P[s_t(a, u) > \hat{\lambda}] \leq \delta - e^{-2mt^2}. \tag{11}$$

Denote the event that that the true value of a function evaluated at point x lies inside the predicted uncertainty ball evaluated at that point, i.e. $u(a)(x) \in C_{\hat{\lambda}}(a)(x)$ as $A(x)$.

$$P[\mathbb{E}_x[A(x)] < 1 - \alpha] = P[\mathbb{E}_x[A(x)] < 1 - \alpha \wedge \frac{1}{m} \sum_{i=1}^{m} \mathbb{1} \{ A(x_i) \} < 1 - \alpha + t]$$

$$+ P[\mathbb{E}_x[A(x)] < 1 - \alpha \wedge \frac{1}{m} \sum_{i=1}^{m} \mathbb{1} \{ A(x_i) \} \geq 1 - \alpha + t]. \tag{12}$$

Note that by calibration we have,

$$P[\mathbb{E}_x[A(x)] < 1 - \alpha \wedge \frac{1}{m} \sum_{i=1}^{m} \mathbb{1} \{ A(x_i) \} < 1 - \alpha + t] \leq P[\frac{1}{m} \sum_{i=1}^{m} \mathbb{1} \{ A(x_i) \} < 1 - \alpha + t] \leq \delta - e^{-2mt^2}, \tag{13}$$

and by Hoeffding bound we have,

$$P[\mathbb{E}_x[A(x)] < 1 - \alpha \wedge \frac{1}{m} \sum_{i=1}^{m} \mathbb{1} \{ A(x_i) \} \geq 1 - \alpha + t] \leq e^{-2mt^2}. \tag{14}$$

Combining above we get $P[\mathbf{E}_x[A(x)] < 1 - \alpha] \leq \delta$.

### 5.2 Mixed Discretization

For the fixed discretization case, instead of sampling $m$ points from the domain for each function in calibration set, we assume $m_i$ points are sampled for each input-output function pair $(a_i, u_i)$. We assume test sample $(a', u')$ has discretization $m'$. The underlying assumption is that even though our query discretization changes, the functions are still exchangeable. Thus, our main analysis still holds, and we just need to account for different discretization errors caused by the varying sampling resolution. The definition of nonconformity score is discretization-dependent, i.e.

$$s_t(a_i, u_i) = \sigma_{\lfloor 1 - \alpha + t \rfloor}$$

where $\sigma_j$ is the j'th smallest value in the set $\{ \frac{\|u(x_i) - \hat{\mathcal{G}}(a)(x_i)\|_2}{E(a)(x_i)} | i = 1, ...m_i \}$. By similar reasoning as inequality 10 in Section 5.1,

$$s_t(a_i, u_i) > \lambda \qquad \Rightarrow \qquad \frac{1}{m_i} \sum_{i=1}^{m_i} \mathbb{1} \{ u(x_i) \in C_\lambda(a)(x_i) \} < 1 - \alpha + t.$$

Since the $t$ term is used in the definition of $\hat{\lambda}$, it also affects the overall relaxation in the quantile value across all functions (of varying discretization). Thus, we need a uniform $t$ across functions, satisfying $t > \sqrt{-\frac{\ln \delta}{2\bar{m}}}$ where $\bar{m} = \min\{m_1, ... m_n, m'\}$. Let $\hat{\lambda} = 1 - \frac{\lceil (n+1)(\delta - e^{-2\bar{m}t^2}) \rceil}{n}$'th quantile of $s(a_i, u_i)$ on calibration set (of size $n$),

$$P[s_t(a, u) > \hat{\lambda}] \leq \delta - e^{-2\bar{m}t^2} \leq \delta - e^{-2m't^2}.$$

By a similar invocation of Hoeffding bound, denote $u(a')(x) \in C_{\hat{\lambda}}(a')(x)$ as $A(x)$,

$$P[\mathbb{E}_x[A(x)] < 1 - \alpha] = P[\mathbb{E}_x[A(x)] < 1 - \alpha \wedge \frac{1}{m'} \sum_{i=1}^{m'} \mathbb{1}\{A(x_i)\} < 1 - \alpha + t]$$

$$+ P[\mathbb{E}_x[A(x)] < 1 - \alpha \wedge \frac{1}{m'} \sum_{i=1}^{m'} \mathbb{1}\{A(x_i)\} \geq 1 - \alpha + t].$$

Note that by calibration we have,

$$P[\mathbb{E}_x[A(x)] < 1 - \alpha \wedge \frac{1}{m'} \sum_{i=1}^{m'} \mathbb{1}\{A(x_i)\} < 1 - \alpha + t]$$

$$\leq P[\frac{1}{m} \sum_{i=1}^{m'} \mathbb{1}\{A(x_i)\} < 1 - \alpha + t] \leq \delta - e^{-2\bar{m}t^2} \leq \delta - e^{-2m't^2},$$

and by Hoeffding bound we have,

$$P[\mathbb{E}_x[A(x)] < 1 - \alpha \wedge \frac{1}{m'} \sum_{i=1}^{m'} \mathbb{1}\{A(x_i)\} \geq 1 - \alpha + t] \leq e^{-2m't^2}.$$

Combining above we get $P[\mathbf{E}_x[A(x)] < 1 - \alpha] \leq \delta$.

## 6 Related Work

**Conformal Prediction:** The conformal prediction framework was introduced in the 2000s (Vovk et al., 2005) to provide distribution-free and finite-sample uncertainty estimates for scalar values. It has subsequently been extended to settings such as regression (Romano et al., 2019), time-series forecasting (Ajroldi et al., 2023), imaging (Angelopoulos et al., 2022), and functional data analysis (Lei et al., 2015; Diquigiovanni et al., 2022; 2021). In this work, we provide a conformal prediction framework for operator learning, and our main distinction from prior work is heteroscedasticity on function spaces. Methods based on modulation or pseudo-density (Lei et al., 2015) may limit learning statistical error patterns across samples and do not adapt uncertainty estimates based on each test sample. We provide heteroscedastic estimates by combining the conformal prediction framework with a base quantile neural operator.

**Approximate Gaussian Methods:** Many existing methods for uncertainty rely on Gaussian assumptions or approximations that result in heuristic, rather than rigorous uncertainty estimates. For example, Magnani et al. (2022) allows for sampling from a Gaussian process to approximate the posterior operator via Laplace approximation, which we have shown to overestimate errors. Akhare et al. (2023) similarly relies on Gaussian assumption and over-estimates uncertainty in empirical studies. (Zou et al., 2023) also uses Gaussian assumption to learn a pointwise noise variance as a heuristic, which does not give a coverage guarantee. By leveraging the conformal prediction framework, we obtain principled uncertainty estimates.

**Data Uncertainty:** Bayesian methods (Zou et al., 2023; Meng et al., 2022) provide the advantage of decomposing uncertainty sources, yet face challenges in scaling up to high-dimension problems. Nehme et al. (2023) studies data uncertainty by learning low-dimensional latent representations of error provides interpretability, but similar faces challenges when an error does not lie in a low-dimensional manifold as

problem dimension grows. Our method focuses on model uncertainty and is complementary to these works. How to incorporate uncertainty decomposition into our method remains an open area for future work.

**Uncertainty in Operator Learning:** As operator learning starts to gain popularity, various works start to explore how to formulate uncertainty in this setting. Guo et al. (2023) explores variance estimation but only provides standard deviation under Gaussian assumption and is constrained to single-point uncertainty estimates. Benitez et al. (2023) approaches uncertainty estimation from a statistical learning lens, and focuses on loss. We provide a bound simultaneously for all points on the domain that is able to capture error structure.

## 7 Conclusion

We show a method of risk-controlling neural operator, which leverages conformal prediction and quantile loss to provide principled uncertainty estimates simultaneously for all points on the domain. We show empirically on a 2D Darcy flow problem and a 3D car pressure prediction problem that our method consistently provides the tightest band while satisfying target calibration percentage among various uncertainty quantification methods. Notably, in the 3D problem, our method is the only method that satisfies target calibration percentage of 98%. We also show that our method helps reveal interpretable error structures via visualization. In addition to empirical results, we also provide proof that shows the calibration percentage guarantee of our method.

While our method is the first introduction of conformal prediction to operator learning, we note that, our method is prescriptive and comes with a principled calibration guarantee, we still rely on a good heuristic uncertainty estimator ($E$ in equation 4) to obtain tight uncertainty balls. In the worst case, if the heuristic estimator $E$ is completely uninformative, our method will yield very wide uncertainty bands. In this work, we focused on expectation over the domain, which is an average, as a measure of risk. Often, other notions of risks, including the worst-case formulation, may be of interest that we leave for future work. There are still many open problems in extending principled uncertainty quantification to scientific ML problems involving PDEs. For example, uncertainty decomposition, better representations of error structure, and efficient sampling are all important directions for future work.

**Author Contributions**

Z. Ma came up with the method, developed the theory, motivated the problem, and performed the experiments. D. Pitt merged the project code with the neural operator library and reproduced the results. K. Azizzadenesheli and A. Anandkumar advised on the overall direction of the project.

**Acknowledgments**

Z. Ma is supported by the Kortschak Scholarship. A.Anandkumar is supported by the Bren Named Chair, Schmidt AI 2050 Senior fellow, and ONR (MURI grant N00014-18-12624). The authors thank Jean Kossaifi for several helpful discussions regarding code reproducibility. The authors thanks Julius Berner, Zongyi Li, and Nikola Kovachki for helpful discussions.

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
