# OpenReview forum: "Calibrated Uncertainty Quantification for Operator Learning via Conformal Prediction"
_TMLR — Accepted by TMLR_

### Review · Reviewer_L1xe · 2024-06-02

**Summary Of Contributions:**

This paper develops a method for predicting the uncertainty of predictions in a PAC style setting, where the goal is to identify a threshold for error alpha, where an 1-delta fraction of new data have error within that threshold.
It achieves this in two steps:
  1.  It uses operating learning (learning a function E) under a quantile regression loss to estimate this threshold for future query point.
  2.  It calibrates this function E, with another parameter lambda, so final estimate of error at any point p is lambda * E(p).  It does this through data splitting.  Evaluating E on a train set, and then using test set to choose lambda to fit the desired (alpha, delta) parameters.

On experiments it does well against some useful baselines.

**Audience:**

Yes

**Claims And Evidence:**

Yes

**Requested Changes:**

Need to address Weakness 4.

I encourage authors to address other weaknesses, as I think will strengthen the paper.  But no strictly required.

**Strengths And Weaknesses:**

Strengths:
 1.  The ideas make sense as presented, and improvements over baselines.  The idea of introducing operator learning allows to harness modern approaches.

 2.  The method does well on the two nicely chosen data sets.

Overall, these probably make the paper suitable for TMLR.

Weaknesses:
 1.  It only considers 2 data sets.  A more thorough paper would have used several more.

 2.  The writing was not that accessible for someone from outside the area due to how it was disjointed.  I think the Theoretical Guarantees in Section 5 may have been better integrated earlier, since some concepts like Split Conformal Prediction were not defined clearly in Section 3 (although Section 5 and Section 4 ultimately can mostly explain what is going on).

 3.  Similarly, this reader would have appreciated more detail on how the operator E was learned.  It cites some papers, but more should be said in the paper.  Was in a NN or a kernel machine or a random forest?

 4.  Figure 6 and 7 overlap.

---

> ### Author Response · Authors · 2024-06-02
> **Addressing review**
>
> We would like to thank reviewer L1xe for finding our work a sensible proposal allowing to harness modern approaches to conformal prediction. Following the reviewers suggestion and comments, we address the raised issues and have uploaded a new version of the pdf.
>
> 1 - Neural operators are function-to-function learning models, and for operator learning studies, we need datasets that are of a functional nature. In this study, we chose Darcy Flow (the first dataset), which is a PDE class widely used in scientific ML, representing problems in material science, fluid dynamics, and reservoir engineering for which uncertainty quantification is crucial for design.
>
> For the second data set, we carefully decided to study the car pressure dataset, as it is a larger-scale, industry-standard dataset recently studied in the context of operator learning [1]. Uncertainty quantification is a heavily sensitive and considered subject in car design, and thus, we use it to test our method to understand uncertainties that emerge from operator learning. Our study on this dataset indicates that the prediction uncertainty is higher in the areas where physics is complex, such as the area on the front of cars, and lower in other areas. We show that our method provides a tight uncertainty bound for the operator learning predictions.
>
> 2 - We expanded on the split conformal formulation in Section 3.
>
> 3 - We elaborate further on E, which is parameterized by the quantile neural operator both in Section 2.3 and 3.1. The core model is a neural operator architecture, for which in our empirical study on Darcy Flow, we utilized the commonly used Fourier Neural Operator (FNO), and for the automotive industry dataset, we utilized the state-of-the-art geometry-informed neural operator (GINO) [1].
>
> 4 - We addressed the overlap between the figure 6 and figure 7.
>
>
>
> [1] Li et al. Geometry-Informed Neural Operator for Large-Scale 3D PDEs, 2023.

---

### Review · Reviewer_7Nsy · 2024-06-10

**Summary Of Contributions:**

This paper proposes a function-space uncertainty formulation for operator learning. This method has better coverage of the function domain and is more scalable. Theoretical results are provided to show the effectiveness of the method together with empirical evaluation showing the method works better than other uncertainty methods such as MCDropout and Laplace approximation.

**Audience:**

Yes

**Claims And Evidence:**

Yes

**Requested Changes:**

Significance:

Can you explain how the alternative approaches are setup to perform well in the experiment setting you use?

Demonstrating scalability:

- To my understanding the bandwidth results are showing the proposed method can perform well compared to others. Can you provide some other metrics on scalability to make it even more clear? For example, is there some larger tasks where it might take a very long time for other methods to tackle but the proposed method can finish in a small amount of time?


Minor issues:

- A missing equation reference at Algorithm 1 line 4, and middle of page 5, end of page 7 and middle of page 10.
- Figure 1: while I think the figure is very nice and interesting, it can be a little difficult to perceive the relative positions of points in 3D. Can you also provide a 2D illustions? (sth like a vertical slice of the landscape). The other thing is it can be hard for colorblind readers to distinguish between e.g. red and green colors, would be nice to enhance the visualized points by using different types of points (dot, cross, triangle for example).
- Page 12 "Equation equation 4"

**Strengths And Weaknesses:**

Strengths:

**Clarity**: overall the paper is clear and easy to follow. The problem is well-motivated. In the intro the authors discussed what is missing in existing methods: function-space coverage, distribution-free calibration guarantee, and scalability. Related works are discussed.

**Novelty**: the proposed method can be considered novel, the theoretical and empirical results are also interesting

**Significance**: Table 1 and Figure 2 show qualitative and quantitative evidence that the proposed method is better than other baselines. Theoretical results also add to significance.

Weaknesses:

**Quality**: overall good quality, but with some typos and missing references

**Significance**: for the alternative methods you compare to, what are the hyperparameter configurations you used? Are these methods giving their best performance? (is the comparison against strong baselines? Or weak baselines)

---

> ### Author Response · Authors · 2024-06-13
> **Addressing review**
>
> We thank reviewer 7Nsy for finding our work clear and novel. Reviewer 7Nsy points out that our method is quantitatively better than baselines and come with theoretical guarantees, which add to its significance. We address the weaknesses below, and we have uploaded an updated version of the pdf based on the comments.
>
> 1. Baseline hyperparameters:
>
> For MCDropout, we did a hyperparameter search which includes varying dropout probability, changing dropout location (projection vs. after each fourier convolutional block), as well as the number of models. For the 2D problem, we found that inserting dropout after each fourier convolutional block with probability of 0.4 gives the best results at 99.5% coverage and 0.00108 bandwidth. For the 3D problem, 0.2 dropout probability provides the best result. Laplace Approximation has no hyperparameters since the approximation is done using the Hessian of the last layer. We use K=3 for neural posterior (increasing K will always increase calibration percentage, but the model complexity will also increase, and thus we need to find a compromise). Deep ensemble has no additional hyperparameters besides model parameters, and we perform normal model tuning. Please also note that, to make these methods applicable to function map learning and relevant to the study of this paper, we generalized them to operator learning.
>
> 2. Demonstrating scalability:
>
> Our 2D model simultaneously provides point uncertainty bands for 421x421=177241 points, and our 3D model provides point uncertainty bands simultaneously 3586 points on the 3D industry level car surface. This scale is unattainable for some prior methods, including [1] which can only output uncertainty band one point at a time (and cannot be aggregated without loosening the bound since points are not independent) and [2] which considers smaller scales of problems with the number of points on the function domain limited to the scale of hundreds.
> We also update Table 1 to include a scalability comparison among the baselines which are possible to run on our datasets. We provide the approximate training time on a single RTX4090 for each method. We keep the model architecture the same across methods, and MCDropout and Deep Ensemble require training 10 models. Our method only requires training 2, the base model and the residual model. Although Laplacian approximation only requires training 1 model, it depends on sampling and obtaining aggregate statistics across samples at inference time. Neural posterior does not require sampling, but does not give a pointwise uncertainty band either - it just outputs the uncertainty (linear) subspace. In the end, please note that, our method is the only one that provides a guaranteed band while other methods come with no guarantee.
>
> 3. Equation references: we fixed them in the updated pdf. Thanks for pointing them out!
>
> 4. Figure 1: we updated Figure 1 with 2D illustrations and different line types (dotted, dashed) in the pdf.
>
> [1] Ling Guo, Hao Wu, Wenwen Zhou, Yan Wang, and Tao Zhou. Ib-uq: Information bottleneck based uncertainty quantification for neural function regression and neural operator learning, 2023.
>
> [2] Xuhui Meng, Liu Yang, Zhiping Mao, José del Águila Ferrandis, and George Em Karniadakis. Learning functional priors and posteriors from data and physics. Journal of Computational Physics, 457:111073, May 2022. ISSN 0021-9991. doi: 10.1016/j.jcp.2022.111073. URL http://dx.doi.org/10.1016/j.jcp.2022.111073

---

### Review · Reviewer_U9P7 · 2024-07-05

**Summary Of Contributions:**

The paper introduces the Risk-Controlling Quantile Neural Operator (UQNO) framework, which leverages conformal prediction to provide calibrated uncertainty estimates for operator learning. This method offers uncertainty quantification over the entire function space, distinguishing it from traditional point-wise approaches. It provides robust, distribution-free, finite-sample calibration guarantees. Empirical validation on 2D Darcy flow and 3D car surface pressure prediction tasks demonstrates that UQNO outperforms existing methods, showing superior performance in both data-rich and data-scarce scenarios.

**Audience:**

Yes

**Claims And Evidence:**

Yes

**Requested Changes:**

1. Compare with More Advanced Methods: Incorporate comparisons with a broader range of recent methods in the field of uncertainty quantification to provide a more comprehensive evaluation.
2. More Experimental Domains: Expand the experiments to include additional application domains to better test the generalizability of the proposed methods.

**Strengths And Weaknesses:**

### Strengths
1. **Interesting Research Problem**: The paper addresses a novel and important problem by applying Conformal Prediction to calibrate operator learning, which is a fresh approach in the field.
2. **Good Performance**: The proposed method demonstrates strong empirical performance, outperforming existing methods in the experiments conducted.
3. **Clear Visualization**: Figure 1 is particularly well-designed and easy to understand, effectively illustrating the core concept of the proposed method.
4. **Theoretical Guarantees**: The paper provides robust theoretical guarantees, enhancing the reliability and rigor of the proposed approach.

### Weaknesses
1. **Lack of Comparison Methods**: While the paper compares with MCDropout, there are many recent methods in the field of uncertainty quantification. A broader comparison with more advanced methods would strengthen the evaluation.
2. **Limited Application Domains**: The method is tested on only two domains. Expanding the experiments to include more diverse domains would better demonstrate the generalizability of the proposed approach.
3. **Heuristic Dependency**: The method relies on a good heuristic uncertainty estimator. If this estimator is uninformative, the resulting uncertainty bands could become excessively wide, reducing the practical utility of the method.

---

> ### Author Response · Authors · 2024-07-06
> **Addressing review**
>
> We thank the reviewer for the thorough review! Reviewer U9P7 identifies our research problem as novel and interesting, and seeing our method’s good performance compared to other baselines, and that we provide theoretical guarantees that other methods do not provide. We address the weaknesses below:
>
> 1. Compare with more advanced methods:
>
> We have added baselines at the request of other reviewers. The latest version has 4 baselines: MCDropout [1], Deep Ensemble [2], and the more recent Laplace Approximation [3], and Neural Posterior Principal Components [4]. We note that our task is uncertainty quantification for functional data, and all the baselines except for Laplace Approximation are designed for finite-dimensional problems.
>
> To compare with finite-dimensional methods we need to first extend them to our functional setting.
> We further highlight that none of the baseline methods provide guaranteed bounds, but our method comes with rigorous guarantees, which are important for safety-critical scenarios such as auto design.
>
> 2. Application domains:
>
> Neural operators are function-to-function learning models, and for operator learning studies, we need datasets that are of a functional nature. In this study, we chose Darcy Flow (the first application), which is a PDE class widely used in scientific ML, representing problems in material science, fluid dynamics, and reservoir engineering for which uncertainty quantification is crucial for design.
> For the second application, we carefully decided to study the car pressure dataset, as it is a larger-scale, industry-standard dataset recently studied in the context of operator learning [5]. Uncertainty quantification is a heavily sensitive and considered subject in car design, and thus, we use it to test our method to understand uncertainties that emerge from operator learning. Our study on this dataset indicates that the prediction uncertainty is higher in the areas where physics is complex, such as the area on the front of cars, and lower in other areas. We show that our method provides a tight uncertainty bound for the operator learning predictions. This application is chosen because it is a challenging real-world setting which requires rigorous uncertainty quantification.
>
> 3. Heuristic dependency:
>
> We do state in the paper that the dependency on a heuristic error estimator is one of the limitations of our method. However, in practice, it is not difficult to train a Fourier Neural Operator (or the same model class as the base estimator) which is good enough as a heuristic error estimator. In fact, in our experiments, we fine-tune the error estimator based on the base estimator, which makes it easy and computationally efficient to obtain this error estimator.
>
> [1] Gal, Yarin, and Zoubin Ghahramani. "Dropout as a bayesian approximation: Representing model uncertainty in deep learning." international conference on machine learning. PMLR, 2016.
>
> [2] Lakshminarayanan, Balaji, Alexander Pritzel, and Charles Blundell. "Simple and scalable predictive uncertainty estimation using deep ensembles." Advances in neural information processing systems 30 (2017).
>
> [3] Magnani, Emilia, et al. "Approximate Bayesian neural operators: Uncertainty quantification for parametric PDEs." arXiv preprint arXiv:2208.01565 (2022).
>
> [4] Nehme, Elias, Omer Yair, and Tomer Michaeli. "Uncertainty quantification via neural posterior principal components." Advances in Neural Information Processing Systems 36 (2024).
>
> [5] Li, Zongyi, et al. "Geometry-informed neural operator for large-scale 3d pdes." Advances in Neural Information Processing Systems 36 (2024).

---

### Decision · Action_Editor_PptP · 2024-08-13

**Recommendation:** Accept as is

**Comment:**

The reviewers all suggest accept, and I agree with them. The paper provides a novel method for uncertainty estimation in complex settings, and the authors provide both theoretical and empirical evidence to justify their claims. Most of the reviewer comments were addressed, except for the request for more datasets, but I do not feel this is reason to not accept it.

**Audience:**

Yes.

**Claims And Evidence:**

The claims made are supported by accurate and convincing evidence. Although some reviewers asked for more domains, I feel the authors chose the domains to highlight challenges their method addresses, so feel it suffices.